# Open-loop analog programmable electrochemical memory array

Peng Chen[1], Fenghao Liu[1], Peng Lin [1,2] ✉, Peihong Li[1], Yu Xiao[1], Bihua Zhang[1] & Gang Pan [1,2] ✉

Emerging memories have been developed as new physical infrastructures for hosting neural networks owing to their low-power analog computing characteristics. However, accurately and efficiently programming devices in an analog-valued array is still largely limited by the intrinsic physical non-idealities of the devices, thus hampering their applications in in-situ training of neural networks. Here, we demonstrate a passive electrochemical memory (ECRAM) array with many important characteristics necessary for accurate analog programming. Different image patterns can be open-loop and serially programmed into our ECRAM array, achieving high programming accuracies without any feedback adjustments. The excellent open-loop analog programmability has led us to in-situ train a bilayer neural network and reached software-like classification accuracy of 99.4% to detect poisonous mushrooms. The training capability is further studied in simulation for large-scale neural networks such as VGG-8. Our results present a new solution for implementing learning functions in an artificial intelligence hardware using emerging memories.

Analog programmability of emerging memories offers a new design paradigm for non-von Neumann computing hardware[1–3]. On one hand, each device in the array can be programmed into thousands of conductance states[4], leading to direct implementation of neural network weights[5–9]. On the other hand, these analog-valued arrays execute matrix multiplications in analog-input-analog-weight fashion[10–12], providing a significant upgrade from a digital implementation in terms of parallelism and computing efficiency[7,12,13]. Despite tremendous progresses in using these emerging device arrays for computing applications[14–17], precisely programming a device towards a specific analog state is still a non-trivial task[18,19]. In practice, emerging devices such as memristors[20–22] are prone to intrinsic switching variations[23–25] and nonlinear conductance change[26,27], stemming from their filamentary switching processes. Consequently, write-and-verify is a commonly used programming scheme[28–31], which employs feedback to guide iterative programming cycles until an acceptable accuracy is attained. However, such a closed-loop tuning process can go beyond 100 cycles[4,12] and contains undesirable operations such as sensing high precision analog-valued conductance. For programming intensive tasks such as training neural networks, closed-loop operation will not only be inefficient, but could also limit the use of array-level parallel programming schemes[32]. As a result, utilizing emerging memories for in-situ learning functions has been greatly limited.

Ideally, analog programming should be an open-loop, linear and symmetric process[18,33]. The conductance change in an analog device, either increasing or decreasing, can be linearly quantized and translated to a stimulating parameter such as a pulse number. This linear and symmetric response[34–37] of the device enables accurate programming from one state to another without the need of verification. Driven by this unique advantage, great efforts have been made in search for a device that could provide the desired open-loop programmability[37–40]. For instance, it was found that the filamentary switching process in a memristor can be regulated by current compliance to produce the desired conductance values[41], which has enabled accurate programming in integrated one-transistor-one-memristor (1T1R) arrays by precisely controlling the drain current of the selecting transistors[41,42].

[1]College of Computer Science and Technology, Zhejiang University, Hangzhou, China. [2]State Key Laboratory of Brain Machine Intelligence, Zhejiang University, Hangzhou, China. ✉e-mail: penglin@zju.edu.cn; gpan@zju.edu.cn

However, this method is more effective in programming a device aiming for an absolute conductance value. In applications such as training neural networks, the weight changes are relative and depend on their present values, which makes the use of current compliance less suitable for the task. In the meantime, ferroelectric memory device (FeRAM) has demonstrated good linear and symmetric programming capability[43], but switching linearity and symmetry is achieved by using voltage pulses with incremental amplitudes, which also adds noticeable complexity in designing the system.

Recently, three-terminal electrochemical memory (ECRAM) devices with a transistor-like structure are proposed for the task. Voltage/current pulses applied at gate terminals drives mobile ions such as $Li^+$, $H^+$ or $O^{2-}$ across the gate stack and effectively modulates the channel conductance through controlled doping/dedoping[44–49]. Linear and symmetric conductance change has been achieved in these ECRAM devices with high reproducibility, fine resolution and good energy efficiency[36,38,50]. Although these observations have generated great expectations for ECRAM in accelerating the training of neural networks, current demonstrations have been more successful in standalone devices, while implementations of ECRAM arrays are still limited by their array size, switching uniformities or functionalities[36,38,51].

In this article, we reported an electrochemical memory array with accurate open-loop analog programmability. Linear and symmetric conductance update of ECRAM was faithfully reproduced in integrated arrays with successful demonstrations in a set of image programming tasks, showing promising programmability for accurate weight update operations. We further experimentally employed our ECRAM arrays for training tasks and in-situ trained a bi-layer neural network to detect poisonous mushrooms with a software-like classification accuracy of 99.4%. In addition to experimental demonstrations, simulation based on device characteristics showed that the ECRAM arrays can achieve highly accurate training of large neural networks such as VGG-8. These results pave the way for developing an efficient and high precision learning hardware for analog and neuromorphic computing.

## Results

### ECRAM array for open-loop programming

ECRAM arrays with $10 \times 10$ array size were fabricated for analog computing (Fig. 1a). We used oxygen-deficient tungsten suboxide ($WO_x$) as tunable channel material, yttria stabilized zirconia (YSZ) as ion conduction layer and tungsten as top gate/reservoir layer for electrochemical gating (Fig. 1b). The YSZ and $WO_x$ layers were partially crystalized during fabrication process which could contribute to better ion conductivity for switching (see Methods and Supplementary Fig. 1). The device exhibited a representative, non-volatile switching hysteresis in current-voltage ($I$–$V$) sweep (Supplementary Fig. 2), and more importantly, the desired linear and symmetric conductance update behavior (Fig. 1c). The conductance of the device could be modulated linearly with different number of up/down pulsing cycles, showing good reproducibility and robustness. The number of identical pulses required to program the device from one state to another can be directly determined by their linear relationship in an open-loop fashion (Fig. 1d), in contrast to conventional write-and-verify method. Similar linear responses can also be achieved using pulse-width modulation (PWM)[52,53] which consolidates a series of identical pulses into a single pulse with variable pulse width (Supplementary Fig. 3). A set of greyscale images of $10 \times 10$ pixels were programmed into the ECRAM array with each pixel tuned by a single PWM pulse without any feedback adjustments (Fig. 1e), showing highly accurate open-loop analog programming capability (See Methods).

### ECRAM characteristics

The open-loop analog programmability of our ECRAM arrays can be attributed to good switching properties of the device. To achieve linear and symmetric response, two physical requirements must be met:

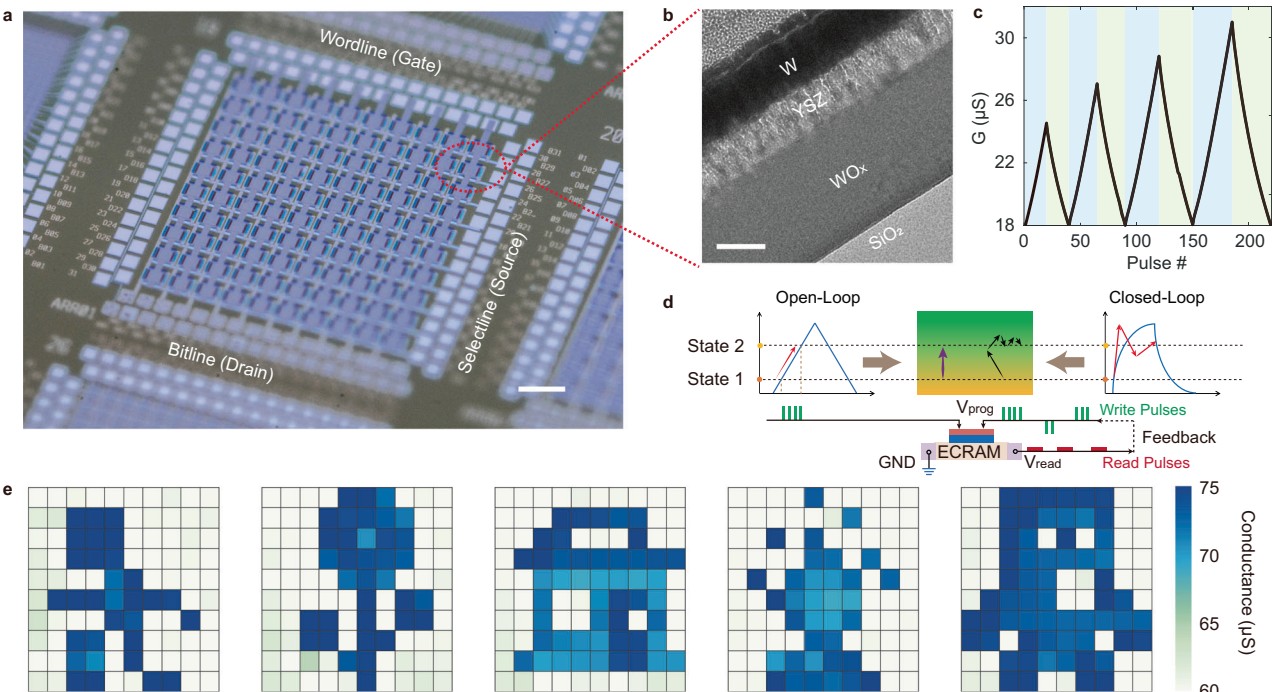

**Fig. 1 | An electrochemical memory array with good open-loop programmability. a** An optical image of an integrated $10 \times 10$ ECRAM array, scale bar: 200 μm. **b** Cross-sectional TEM micrograph of the gate stack (W/YSZ/$WO_x$), scale bar: 50 nm. **c** Demonstration of linear and symmetric conductance update in an ECRAM using a series of up/down pulsing cycles of different pulse numbers (20, 25, 30, 35 pulses, ±5 V, 150 ms). **d** Schematic of open-loop and closed-loop programming with inherently different pulsing sequences. **e** Programming of five image patterns in a $10 \times 10$ array, all image pixels were programmed using open-loop scheme and single PWM pulses.

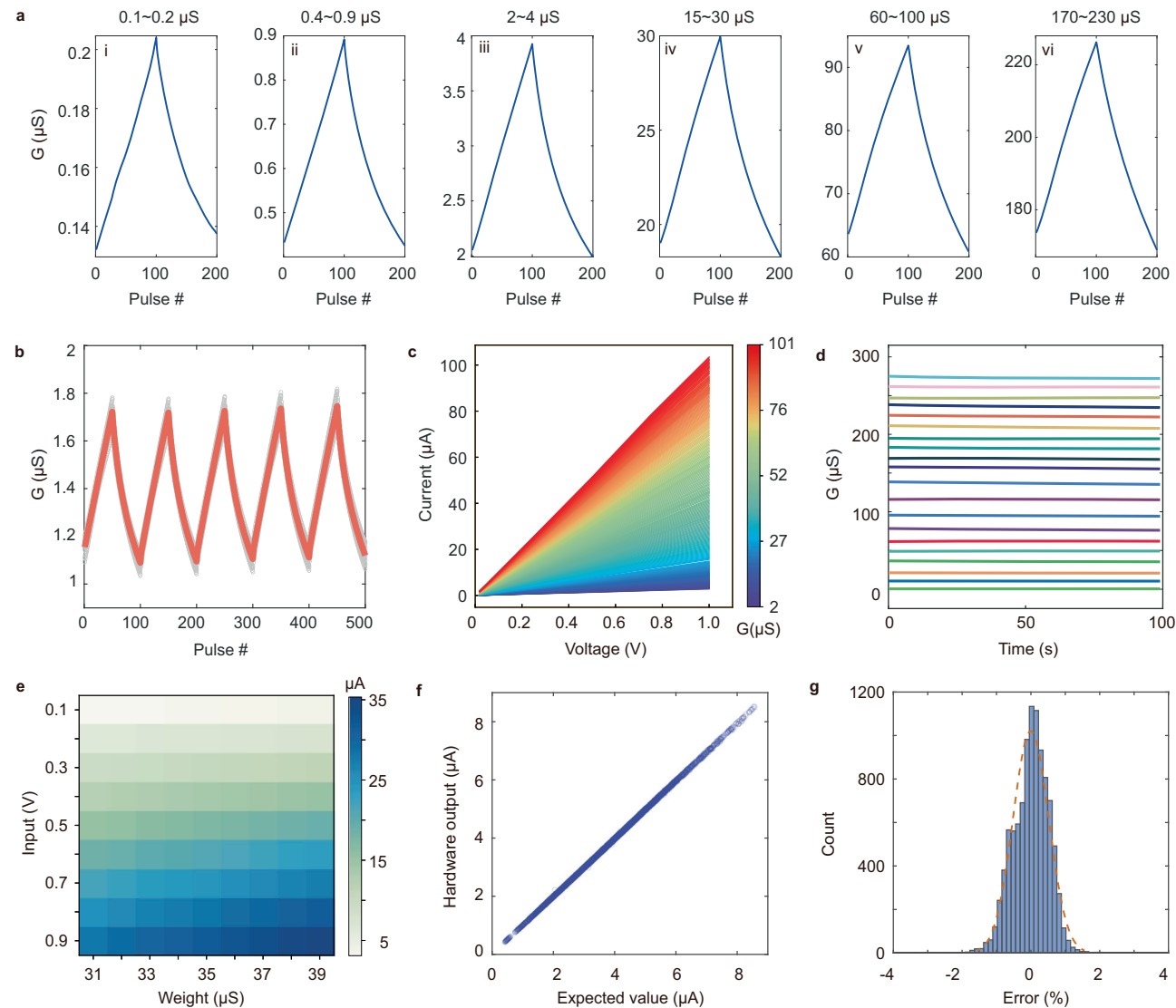

**Fig. 2 | Characterizations of the ECRAM. a** Linear and symmetric conductance update across different dynamic ranges of (i) 0.1 - 0.2 μS, (ii) 0.4 - 1.0 μS, (iii) 2 - 4 μS, (iv) 15 - 30 μS, (v) 60 - 100 μS and (vi) 170 - 230 μS, each cycle consists of 100 up/down voltage pulses (± 5 V/100 ms). **b** Superimposed plot of cycling tests of 10 ECRAM devices over 5 repeated cycles (±5 V/100 ms), showing good uniformity. **c** *I*–*V* plots of an ECRAM with 256 monotonically potentiated states, demonstrating good **I**–**V** linearity and wide range programmability. **d** Retention test of 20 analog states over a wide switching window. **e** Analog-input-analog-weight multiplications between different pairs of device conductance and input voltages. The uniform color gradient of output current levels indicates good computing accuracies. **f** Statistics of experimental VMM in the 10 × 10 ECRAM array from 1000 random input voltage vectors. The experimental outputs are in a good agreement with expected values. **g** Histogram plot of the output error from the VMM.

(1) a quasi-linear relationship between the conductivity and oxygen concentration of the $WO_x$ channel[54], and (2) a stable ion injection/ejection rate at $WO_x$/YSZ interface during programming. Theoretically, the use of current source offers better control over the current injection rate[35,51]. Nonetheless, we chose voltage source for this work owing to its simplicity for array implementation[51]. The device structure and material compositions were carefully optimized through fine-tuned fabrication processes (see Methods), and different channel conductance were explored for quasi-linear responses. It was found that our ECRAM device exhibited linear and symmetric behavior in widely-spread switching windows, ranging from sub-μS to hundreds of μS (Fig. 2a). We calculated the asymmetry non-linearity (ANL) factor[55] of these conductance windows to be 0.21 ± 0.05 (Supplementary Fig. 4), which is significantly lower than those reported for memristors[55]. For a continuous sweep over a large dynamic range, the linearity of conductance update was slightly degraded (Supplementary Fig. 5). However, it was found that maintaining switching symmetry is sufficient to

guarantee good convergence in training neural networks while linearity mainly affects the effective learning rate of each weight update[18,32,56]. Meanwhile, the metrics of a discrete device could be further relaxed by modifying training algorithms[32].

In addition, we found that the linear and symmetric update in our ECRAM is highly reproducible and uniform from device-to-device (D2D) and cycle-to-cycle (C2C). To assess the uniformity of the switching behavior, a cycling test consisting of 50 potentiation/depression pulses (±5 V, 100 ms) were repeated among 10 devices for 5 cycles (Fig. 2b), exhibiting a low spatial-temporal variation of 2.3% ($\sigma$/$\mu$). The same test could be repeated for >40 consecutive up/down cycles with consistent linearity and symmetry (Supplementary Fig. 6) and maintaining good switching properties after >50 million pulse stimulations (Supplementary Fig. 7).

While good programmability of ECRAM is highly favorable, in-memory computing functions such as multilevel storage and matrix multiplications are also required for training neural networks.

Figure 2c shows an I-V plot of an ECRAM device with 256 states gradually potentiated by 256 set pulses. The conductance of the device increased monotonically and uniformly. A small voltage sweep (0–1 V) was applied after each pulse to readout the programmed state, which shows linear I-V conduction between the source and drain contact. Additional data retention tests further confirmed the stability of the programmed analog states ranging from sub-μS to over 200 μS (Fig. 2d). The linear I-V relationship and multilevel storage capability were previously demonstrated in oxide memristors, which are highly desired properties for analog-analog matrix-multiplications[28]. Figure 2e shows the analog-analog multiplication capability of our ECRAM array, in which a column of nine ECRAM devices were programed into equally-spaced conductance states of 31, 32, 33, …, 39 μS, and subsequently multiplied by the input voltage vector of 0.1,0.2, 0.3, …, 0.9 V. The output current of each input-weight pair was plotted in the figure and shows a uniform color gradient. Vector-matrix multiplication (VMM) in a 10 × 10 ECRAM array was also evaluated. During read and VMM operations, input voltage pulses were applied at the source terminals through the select line electrodes, and the current were readout at the bit line outputs (Supplementary Fig. 8). 1000 randomly-generated binary input voltage vectors were applied serially to the ECRAM array with randomly initiated device conductance. The VMM results were readout and analyzed, which were in a good agreement with the expected results (Fig. 2f) and demonstrated low computing errors (Fig. 2g).

## Robust analog programmability in ECRAM array

For training neural networks, the weights are directly stored in memory arrays and updated through repeated training cycles. The accuracy of the neural network is determined collectively from the update precision of each device. Therefore, reproducible analog programming from an entire array is highly important. Figure 3 shows a series of open-loop programming tasks performed in our ECRAM array. Each device is updated from its present conductance state to a target state using a single voltage PWM pulse with pulsewidth solely determined from its switching linearity (see methods). Two 10 × 10 gradient patterns with different orientations were chosen for the task (Fig. 3a). A programming cycle to switch between these two gradient patterns can comprehensively evaluate the analog programmability of our ECRAM array because weight changes of the 100 pixels uniformly cover different update amplitudes and polarities (see insets of the conductance update matrix in Fig. 3a). Figure 3b shows the experimental results of the programming process. The two gradient patterns were serially programmed into the array from a random initial state, and could be programmed back and forth for multiple cycles, all using the open-loop PWM scheme. Notably, a single PWM pulse cycle was sufficient to program a new gradient pattern from the previous one, showing good spatial uniformity of the analog programming process. By iteratively switching between two patterns, we further demonstrated good reproducibility and robustness of our ECRAM in carrying out these programming tasks. Figure 3c shows the program errors of each device in the cascaded programming cycles. Most of the program errors fell below 3% (for 90% of the devices) and 1uS (Supplementary Fig. 9), and the average write error was calculated to be 1.07% (Fig. 3c), defined as the ratio of conductance mismatch over target values. The programming accuracy of our array using open-loop scheme is already comparable with memristor arrays tuned by dozens of write-and-verify operations[12].

For tasks demanding even higher programming accuracy, such as for data storage or downloading weights from an ex-situ trained neural network, feedback may be used to further reduce the programming errors. Figure 3d shows the evolution of programmed pattern in three consecutive pulsing cycles. 99 out of 100 devices have achieved programming error <1 μS after three cycles (Supplementary Fig. 10). The average conductance mismatch $|G_{actual} - G_{target}|$ between the

programmed pattern and the target values were reduced from 0.46 μS to 0.21 μS in the 2nd cycle, and further lowered to 0.16 μS after three cycles, achieving high programming accuracy with average programming error of only 0.41%. It shows that the linear response of the ECRAM devices was also beneficial in closed-loop programming scheme, providing fast convergence ability during iterative write-and-verify cycles.

## Multilayer neural network training with ECRAM array

Combining the in-memory computing capability demonstrated in Fig. 2 and reproducible analog programming capability in Fig. 3, our ECRAM array provides a promising toolset for in-situ training tasks. To show that our ECRAM can be used for accurate learning tasks, we trained a 10 × 5 × 2 bi-layer neural network to separate poisonous mushrooms from esculent ones, using a database taken from The Audubon Society Field Guide to North American Mushrooms, available at UCI machine learning repository[57] (Fig. 4a). The database contains 8124 samples of mushrooms (poisonous: edible = 48.2%:51.8%), characterized by 22 attributes scored from 0 to 11. We used the first 10 attributes (e.g., cap-shape, odor, gill-size, etc.) to train our network (Fig. 4b). The 10 × 10 ECRAM array was partitioned to host two neural network layers, which were placed in different columns using a total of 60 devices (Fig. 4c). Each signed weight was mapped to a single ECRAM cell by an offset (see Methods). For hardware implementation, only synaptic functions have been implemented in arrays, while the processing of activation function, routing data between first and second neural network layers, as well as the calculation of gradients for back-propagation were done in software. The bilayer network was trained by backpropagation to minimize the mean square error (MSE). A single PWM pulse was applied to program each device for the weight update by using a global weight-to-pulsewidth coefficient. After the in-situ training, the bi-layer neural network reaches a classification accuracy of 99.4% on the whole database (Fig. 4d). To understand how our ECRAM performed during the training process, we recorded the conductance evolution of the 60 devices during 70 training epochs (Fig. 4e) and calculated the programming errors for each update (Fig. 4f). Most devices have shown small program error within 1 μS through the training process, thanks to the linear and symmetric responses. Since we used a global mapping between the weight and pulsewidth, some devices have exhibited relatively larger errors than the others, due to spatial variations of the devices. However, the neural network and its in-situ training process is highly adaptive to small percentage of non-ideal cells, as we noticed that the programming errors were gradually minimized during the training process (Fig. 4g). Overall, the ECRAM array has demonstrated highly accurate and robust analog programming capability for the demonstrated training task.

The performance of our ECRAM in training large-scale neural networks was further evaluated by simulation using VGG-8 network and CIFAR-10 dataset[58] (Fig. 5a). The convolutional neural network consists of 3 convolutional blocks and 2 fully connected layers with total of 8 network layers. The software-based weight update processes were modified to include the impact of device nonlinearity, spatial and temporal variations, all extracted from experimental data (see Methods). The simulation based on device characteristics achieved a classification accuracy of 87.1% after 175 epochs, which is close to ideal device with perfect linearity, symmetry and uniformity. For simulation conditions with increased variability, gradual degradations in classification accuracies were observed, suggesting that securing uniformity and reducing outlier devices may be important in the training process of larger neural networks.

In summary, we reported a highly functional ECRAM array for efficient pattern programming and in-situ training of neural networks. Open-loop analog programmability in array-level demonstrations was achieved utilizing the intrinsic device properties such as good

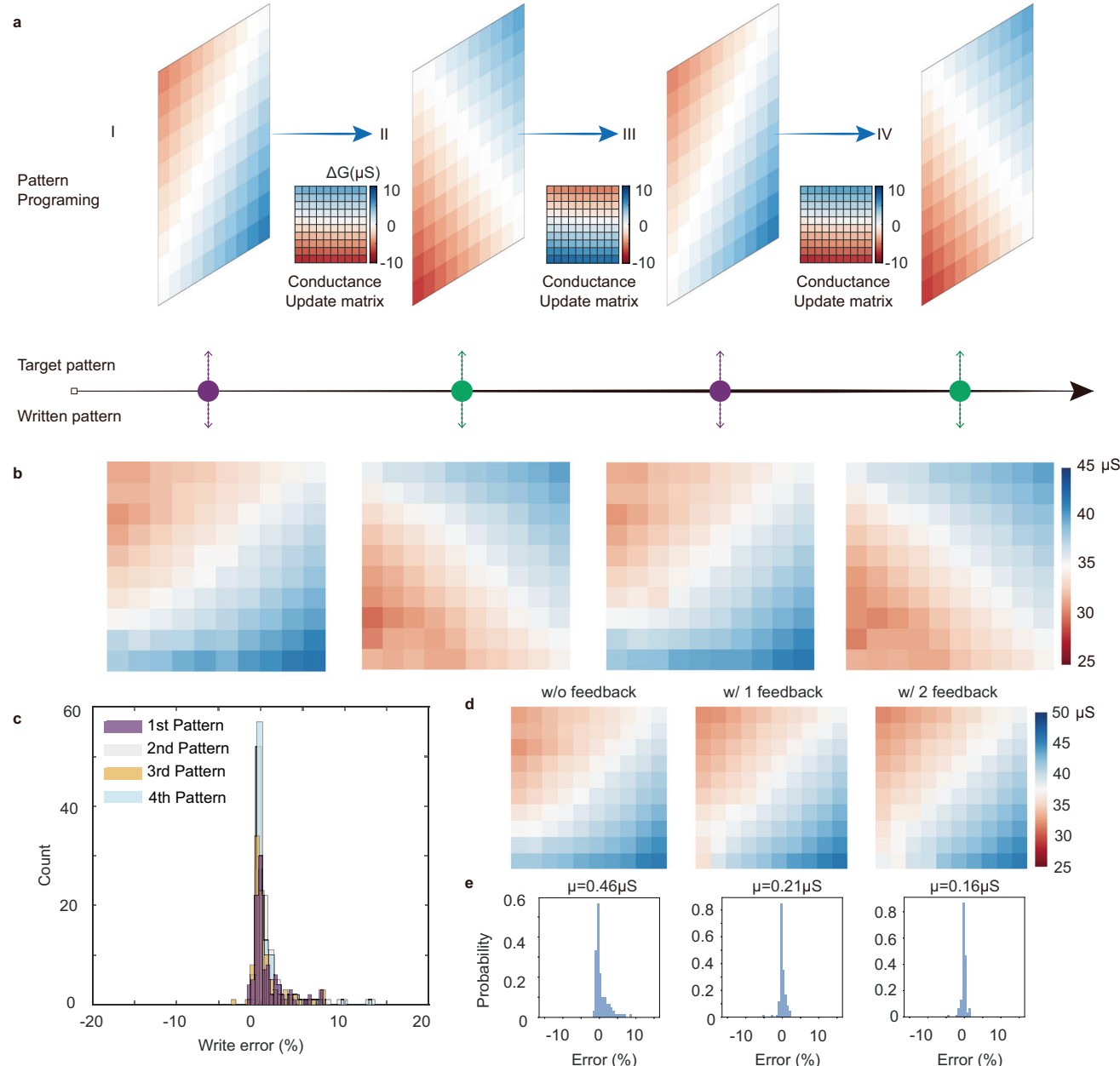

**Fig. 3 | Continuous open-loop pattern programming in a 10 × 10 ECRAM array.**
**a** Timeline illustration of array programming process. Two patterns with different gradient orientations were programmed back and forth into a 10 × 10 array. Targeted conductance changes of the devices were uniformly distributed from different amplitudes and polarities to comprehensively evaluate the programming capability. **b** Experimental results of continuous open-loop programming of gradient patterns using single PWM pulses, good programming accuracy was achieved without any feedback adjustments. **c** Distribution of write error for four open-loop programming cycles. The relative write error is defined as $(G_{actual} - G_{target})/G_{target} \times 100\%$. **d** Pattern programming using write-and-verify scheme, for applications that demand even higher programming accuracies. **e** Distribution of write errors for each write-and-verify cycle.

spatial-temporal uniformity, programming linearity and symmetry. Training neural networks with open-loop analog programming in our ECRAM array can significantly reduce the design complexity of the system, which is highly advantageous over existing write-and-verify methods. The demonstrated analog programmability of our ECRAM have contributed to software-comparable training accuracy for neural networks. The efficacy of the ECRAM array may be further improved in future studies by optimizing switching speed and reducing outlier devices in large-scale integrations. The CMOS compatibility of the ECRAM array should also be addressed in integrated systems, ensuring consistent device performance undergoing BEOL fabrication process and low voltage operations matching CMOS designs in advanced technology nodes. In the new era of artificial intelligence which

trending for large language models (LLMs), development of highly functional learning hardware for training neural networks could be even more important. Our ECRAM may fill in the gaps of current in-memory computing technologies with excellent programming and learning capabilities.

## Methods

### ECRAM array fabrication

ECRAM arrays were fabricated on top of a silicon substrate with a 100 nm thick $SiO_2$ layer. The fabrication process is schematically illustrated in Supplementary Fig. 11. First, Ti (3 nm) / Pt (40 nm) selectlines were formed by photolithography and DC sputtering. A 100 nm $SiO_2$ layer was then deposited using PECVD to provide isolation between

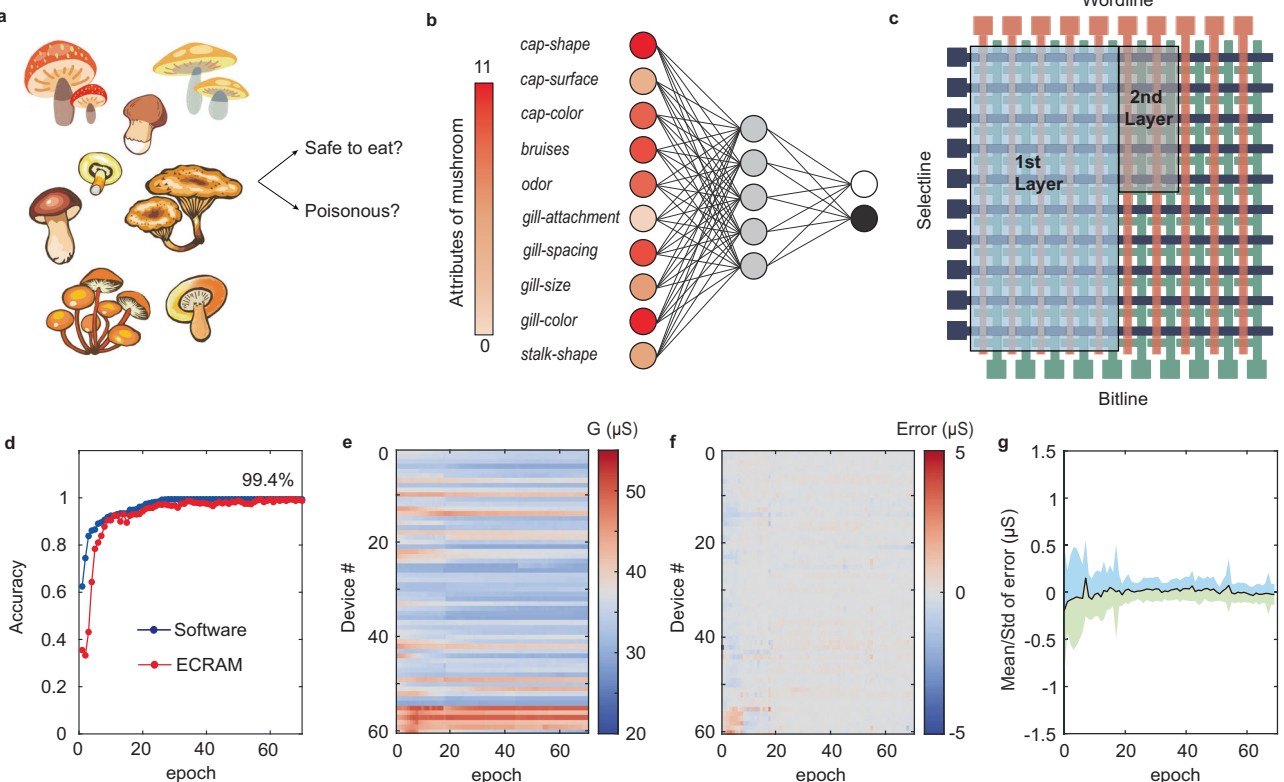

**Fig. 4 | In-situ training using ECRAM array. a** Schematic illustrations of edibility detection on a mushroom dataset, **b** Configuration of employed bi-layer neural network. First 10 attributes out of 22 from the mushroom database were adopted for training and classification, and mushroom samples were classified into edible and poisonous. **c** Schematic illustration of weights mapping into an ECRAM array, the 1st layer of 10 × 5 network size was placed in the left 5 columns and the 2nd layer of 5 × 2 weights were further assigned to another two columns. Each cell represents a signed weight by an offset. **d** Accuracy evolution curves as a function of training epochs derived from hardware incorporated neural networks, achieving classification accuracy of 99.4%, which matched the accuracy of pure software training (blue). **e** Evolution of the conductance and **f** programming errors of the 60 devices through 70 training epochs. **g** Statistical analysis of programming error through training, the colored area shows the margin between μ − σ and μ + σ.

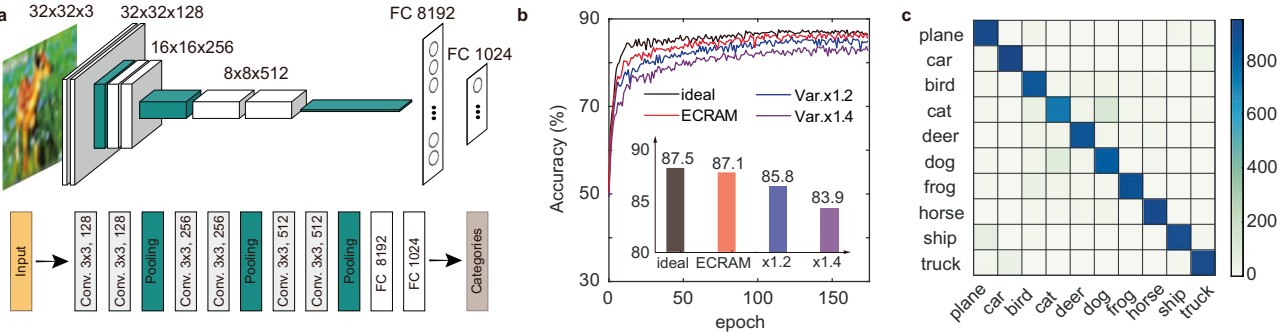

**Fig. 5 | Simulation of VGG-8 with CIFAR-10 database. a** The network structure of the VGG-8 network for CIFAR−10 image classification. **b** Simulated training in ECRAM array with different level of device variability. **c** The confusion matrix for trained VGG-8 on the test set.

bitlines and selectlines, followed by photolithography and ICP-RIE process to selectively expose bottom selectlines electrodes for electric contact to source terminals. Ti (3 nm) / Pt (40 nm) / W (5 nm) stack bitlines and source contacts were formed at the same time by photolithography and DC sputtering. After that, 100 nm thick $WO_x$ channel were patterned and deposited using reactive DC sputtering with a W target under the Ar / $O_2$ (4:3) mixing atmosphere. The sputtering power density was set to 4.5 W/cm². A rapid thermal process at 400 °C for 30 s was subsequently conducted for modulating the crystallinity and electrical properties of obtained $WO_x$ channels. 50 nm thick 8 wt. % YSZ electrolyte layer was further formed by RF sputtering under Ar ambient. Both $WO_x$ and YSZ layer were patterned by photolithography for good

electrical isolation between different cells in the array. Finally, wordlines of 50 nm tungsten were formed by photolithography and DC sputtering to complete the array fabrication process. All patterns, including metal electrodes, $WO_x$ channel and YSZ layers, were formed using lift-off process. The overall fabrication process was designed to avoid excessive thermal and chemical exposures that could degrade the device performance, which should be further optimized for the BEOL process to build an integrated ECRAM chip.

## Characterizations
The cross-section of the device was captured using a transmission electron microscope (FEI Tecnai G2 F20 S-TWIN) after focus ion beam

thinning to a thickness of 100 nm (FEI Quanta3D FEG). Electrical measurements of ECRAM device were mainly conducted at ambient temperature by using semiconductor parameters analyzer (B1500A, Keysight) equipped with functional pulse measurement unit (B1530A). The DC transfer characteristic curves were recorded by scanning gate voltage while monitoring the source and drain with a readout voltage of 0.1 V. Pulse measurement was carried out by applying a sequence of positive and negative voltage pulses to the device, then the conductance changes were obtained by sampling the recorded source-drain current with a read pulse. The evaluation of the programming speed of the ECRAM should not only consider the pulsewidth of the write pulse, but also the amount of conductance change and signal/noise ratio. The settling time between the write and read pulses, i.e. the read-after-write delay, should also be considered when evaluating the overall programming speed of ECRAM[59]. For the up/down test, the time interval between write and read pulses was set to 100 ms. The Device-to-device variation is calculated from 10 devices, in which all states including potentiation and depression (5 cycles) are aligned according to pulse number, then the average conductance ($\mu$) and standard variation ($\sigma$) of each state is calculated for evaluation of average variation ($\sigma/\mu$). For reading the device's conductance using DC sweep, we used sweep voltages from 0 to 1 V across the source and drain electrodes at a sweep rate of 0.8 V/s.

### Array measurement

The functionality of the ECRAM array was tested using a PCB testing board combined with a probe card, which can apply voltage pulses to one column or device cell and measure currents (Supplementary Fig. 12). The fabricated arrays were connected to the test system through a 128 pins probe card. The testing system allows addressing and operating an array with customized Python programs. The testing system is controlled by a series of Python scripts. For array programming, the ECRAM devices were updated in series using 1/2 V scheme to mitigate crosstalk in the array, utilizing the nonlinear relationship between the conductance change and programming voltage amplitudes (Supplementary Fig. 13 and 14). Single pulses with variable pulse durations were used to change devices from one conductance state to the other. We first estimated the unit conductance change ($\Delta G_{min}$) per 1 ms pulse, and which value was adopted as a global parameter for all devices to calculate the pulse duration of each update. For open-loop programming, a single programming pulse was applied to each selected device without feedback, while for closed-loop programming, the device was programmed using iterative read-program-read cycles. For reading operations, the word lines (WL), as well as all unselected bit lines (BL) and select lines (SL) were grounded (Supplementary Fig. 8). Read voltages of 0.5 V were applied at the SLs while current was readout at selected BL.

### Hardware incorporated neural network training

For bilayer neural network training in ECRAM array, we employed softmax activation for both the first and second layers, and the activations were implemented in software. The first 10 attributes of the mushroom dataset were selected as the input parameters to the bilayer neural network, as summarized in Supplementary Table 1. Different descriptions of each input parameter were encoded numerically (e.g., different Cap-shapes were encoded from 0 to 5). The conductance ($G$) of all used crosspoints devices were initialized with a linear mapping relationship with weight ($w$): $w = (G - G_{min})/(G_{max} - G_{min}) - $ offset, where $G_{max}$ and $G_{min}$ represent the conductance window we used for training, and the offset can be done either in software or using a reference column in the array. The weights (from -1 to 1) were linearly mapped to conductance ranges between 25 μS and 50 μS, which provides fine granularity for weight updates and good signal-to-noise ratio for inference tasks with around 200 resolvable steps using a write pulse width of 100 ms. This conductance range was chosen empirically, which provided us with consistent training results for current

task. During the training process, we employed the MSE error back-propagation algorithm to generate weight gradients in the neural network. After the forward process in each epoch, the neural network loss and weight gradients were calculated in software, and linearly mapped to required conductance change. Then the ECRAM array were updated according to derived gradients for each training epoch. Here, single pulse with calculated pulse widths was employed element-wisely update all devices in the hardware neural network.

### Simulation of large neural network

The simulation is conducted based on the Python framework of 'DNN +NeuroSim'[58]. The VGG-8 network consists of 3 convolutional blocks and 2 fully connected layers with total of 8 network layers, which was trained with CIFAR-10 dataset. After training with 50,000 images, the neural network is tested with 10,000 images for benchmarks of different devices models. Behavioral device model of ECRAM was built by fitting the experimental data shown in Fig. 2b. Cycle-to-cycle variation of 0.8% and device-to-device variation of 2.3% was used for the simulation. To provide a coarse estimation of outlier devices in large-scale integrations, additional simulation with enlarged variation value of (×1.2, ×1.4) was performed. The large variation ratio was applied to both device-to-device and cycle-to-cycle variations, while keeping other parameters unchanged.

## Data availability

The data that support the plots presented in this paper as well as other findings derived from this study are available from the corresponding author upon reasonable request.

## Code availability

The codes used in this paper are available from the corresponding author upon reasonable request.

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

## Acknowledgements

This work was supported in part by STI 2030 Major Projects (2021ZD0200400), Natural Science Foundation of China (61925603), The Key Research and Development Program of Zhejiang Province in China (2020C03004), Major Program of Natural Science Foundation of Zhejiang Province in China (LDQ23F040001), Fundamental Research Funds for the Central Universities.

## Author contributions

P.Lin. conceived the idea. P.Lin. and G.P. supervised the team. P.C., P.Lin. and G.P. designed the experiments and prepared the manuscript. P.C. fabricated the devices and conducted electrical measurements. F.L., P.Li, P.C. and B.Z performed algorithm study and simulation. P.C., Y.X. and F.L. performed the array measurement. All the authors contributed to the discussion and analysis of the results regarding the manuscript at all stages.

## Competing interests

The authors declare no competing interests.
