## [Peer Review File · Nature Communications]

REVIEWER COMMENTS

Reviewer #1 (Remarks to the Author):

In this manuscript entitled “Open-Loop Analog Programmable Electrochemical Memory Array”, the authors demonstrated an ECRAM array for image storage and in situ training of neural networks. The ECM can show good performance in terms of various parameters. Overall, the paper is with high quality and the results are interesting and important. I would recommend its publication, given that some minor issues are addressed.

1. The reported ECRAM has shown very promising device performance, while the material characterizations were limited. It would be very helpful to provide some elemental analysis, such as the composition of the channel material.
2. In fig. 2, it seems that the computing was only shown for multiplication, rather than multiply-accumulate operation that has been demonstrated in RRAM arrays. Would the author comment on that?
3. The authors showed that the device can achieve linear update in several different conductance ranges, while the retention data was only tested for 20-60 μs . It may be worth knowing how the device retention looks like in other switching windows, such as sub- μs and over 200 μs ?
4. The open loop programming results are very impressive. Usually in RRAM arrays there are always some defective devices. I understand that ECRAM is non filamentary switching which means it may be more reliable, but can author comment on the reproducibility of the reported ECRAM?
5. The authors have only shown schematic of device fabrication, it would be better to show how the array was fabricated.
6. There is a typo in the caption of fig. 5, probably some missing words from editing.

Reviewer #2 (Remarks to the Author):

The submitted manuscript provides a demonstration of the unique properties and potential of Electrochemical Random Access Memories (ECRAM) as artificial synapses for in-memory compute.

However, this reviewer requires further details and clarifications to properly gauge the novelty and impact of this systematic work. If you could please help understand and address, where applicable, the following.

1. The manuscript puts an emphasis on "open-loop" programming and claims it is superior to the burden added by a write-verify scheme. That is only truly differentiating if it is combined with massively parallel and non-serialized programming. Is this report using a parallel programming scheme in the array demonstrators where all devices are updated simultaneously? If yes, please provide the details of such a scheme, and a plot showing how each device in an array can be programmed in such scheme with limited impact associated with half-select of its nearest neighbors.

2. The manuscript demonstrates a 10x10 array with impressive programming range, including at the relatively high channel resistance necessary to scale to larger array while maintaining signal to noise. However, there is no diagram or clear description of the metal lines topology. In particular, the "wordlines" are said to be deposited subsequently to electrolytes. Are the active areas of devices defined in this case, or are the WO_x & YSZ deposited as blankets? If the latter, how can you electrically isolate devices if WO_x is not patterned?

3. The manuscript repeatedly mention the need for linear characteristics, but for deep learning applications, that is not a requirement, what should be emphasized is the symmetry allowing for algorithms to converge when used in learning tasks. In addition, it should be stated that modifying training algorithms can relax the metrics a discrete device needs to achieve. (see Gokmen and Haensch, Algorithm for Training Neural Networks on Resistive Device Arrays, Front. Neurosci., 26 February 2020)

4. A series of demonstrations are provided using the array: pattern transfer, weight transfer, and "in-situ" training. A general observation is that for inference tasks (such as the first 2 demonstrations) the use of ECRAM can hardly be justified over established technologies such as PCM, RRAM, Flash, etc... since you would transfer information once and then spend the vast majority of time and energy performing matrix-vector multiplication, making use of serial write-verify acceptable. Moreover, for the weight transfer demonstration, the network can be trained in software using "hardware-aware" noise injection to make it more resilient to known devices non-idealities and degradation. Because of the unique deterministic and controllable properties of ECRAM, it is the recommendation of this reviewer to focus on the training demonstration. Indeed, this is where it provides true differentiation. This could also allow the authors to provide more details on that particular achievement.

5. It appears that the favored programming voltages and pulse durations are +/-5V and 100ms, respectively. What are the thoughts of the authors on how to reduce both numbers, especially pulse width, without degrading signal to noise ratio?

6. What is the authors' expectation on the impact of a BEOL integration process environment on the array properties during steps following active area formation? (exposure to further thermal budget, chemistry, ionic species, moisture, etc...)

This reviewer hopes that the above will be of use to the authors and can be used to further amplify the systematic study of their ECRAM devices + arrays, and complement the unique demonstrations presented.

John Rozen

Reviewer #3 (Remarks to the Author):

In the present manuscript, the authors disclose a study on the open-loop programming of electrochemical memory arrays. The authors demonstrate that the devices can be programmed highly accurately using a pulse-width modulation scheme. The devices show very linear and symmetric tuning which makes them suitable as weights in online learning systems using backpropagation. In addition, the authors demonstrate the integration of the device in a small array as well as the programming within the array. In-situ training is demonstrated on the array, mainly by programming the respective devices. Based on the device's properties a large-scale network is simulated using standard software. ECRAMs have been studied by different groups. Also, small arrays have been demonstrated (e.g., by Cui et al, Nature 2023, ref. 47 in the manuscript). Thus, the device concept itself is not novel, also the realization of an integrated array has been demonstrated before. The novelty is shown in the programming scheme. On the experimental basis shown in the manuscript, one cannot draw the conclusion of superior properties. A lot of information is missing to be able to reproduce the results by the authors. The manuscript requires extensive revisions according to the comments outlined below.

1) While the programming scheme and properties look very promising the study does not prove that the integration of large arrays with low variability is feasible. The authors demonstrate here a very low d2d variability for 10 devices over 5 cycles. For the network shown in Fig. 5, however, a lot higher number of devices is required, and the devices will be trained for more than 5 cycles. Even though most of the devices work well, the outliers will limit the overall performance. This might be 1 out of 1000 or even less devices, but for large scale networks this needs to be guaranteed. Such an

investigation is certainly out of the scope of this paper, but one should be careful to claim superior properties with such a low number of statistics.

2) What is the endurance of the devices? Some thousands of cycles are shown, but what is the limit?

3) For the programming the authors use pulses of 5V and 150 ms. The voltage is too high to be integrated with modern CMOS. Thus, special transistors and drivers would need to be used creating circuitry overhead and additional power consumption. The pulses are quite long with 150 ms, leading to slow speed operation and a higher energy consumption.

4) The programming of the device in the array is not well explained. The ECRAM is a three-terminal device. The lines in the array are not labeled. What are the gate lines, and which are the drain and source lines? According to the supplement, the authors connect all pins of the array to a probe card. What signals are applied to all lines when programming? Do you program one cell at a time? How is program-disturb avoided in the other cells sharing the same gate line?

5) What is the read voltage used? How is the read performed in the array, when reading a single cell and a whole line for doing the VMM operation?

6) The multiplication test in 2f is not well explained. In general, one should program random patterns to the devices and use random inputs, but it is not clear if this has been done. Here, it is also helpful to discuss the addressing.

7) On page 6, the authors mention that a voltage sweep is used to read the device state. What was the used sweep rate?

8) For the demonstration of the two-layer network, the authors store the weights for both layers in the same array. It is not clear how inference and backpropagation is realized here. How is the output of the first layer introduced in the 2nd layer? As there is no additional circuitry, the authors must do this in software, but the explanation is lacking. Which activation function is used between the first and second layer? The output of the VMM in the first layer is current, which needs to be transferred to a reading voltage pattern. The reading needs to be done sequentially here.

9) In Fig. 4d, the accuracy of the network is demonstrated, but a saturation has not been achieved. Why did the authors stop after about 38 epochs. What happens further on? The authors should train until a saturation is achieved as for example in Fig. 5. For benchmarking it would be good to show the accuracy that is expected using a pure software approach.

10) For the large-scale network the authors used a simulation tool. Which software has been used? Which device characteristics for the update have been used? Did the authors include the observe variability in the training? As mentioned in my first comment, it is not proven that the variability is as low for the high number of devices required for the simulated network. The used software also does not consider losses due to the CMOS circuitry and the analog to digital conversion.

AUTHORS' RESPONSE TO THE REVIEWER'S COMMENTS

We thank the editor and reviewers for their precious time and valuable feedback that have helped us to improve the quality of our manuscript. We have thoroughly revised the manuscript and made substantial changes to address the reviewer's comments. The point-to-point responses are as follows.

Reviewer #1 (Remarks to the Author):

In this manuscript entitled “Open-Loop Analog Programmable Electrochemical Memory Array”, the authors demonstrated an ECRAM array for image storage and in situ training of neural networks. The ECM can show good performance in terms of various parameters. Overall, the paper is with high quality and the results are interesting and important. I would recommend its publication, given that some minor issues are addressed.

RESPONSE: We sincerely thank the reviewer for the positive comments on the novelty and significance of our work. We appreciate the reviewer for the constructive suggestions to improve our manuscript.

1. The reported ECRAM has shown very promising device performance, while the material characterizations were limited. It would be very helpful to provide some elemental analysis, such as the composition of the channel material.

RESPONSE: We thank the reviewer for this constructive suggestion. In the revised manuscript, we used HRTEM/SAED to check the pristine YSZ/ WO_x interface. The SAED data revealed that both YSZ electrolyte and WO_x channel were partially crystallized, which could contribute to better ions conductivity for switching. We added these results as Supplementary Fig. 1 to SI.

Supplementary Figure 1. Material characterizations of pristine YSZ/ WO_x interface. a, High resolution transmission electron microscope (HRTEM) image of YSZ/ WO_x interface. b, Selected area electron diffraction (SAED) pattern shows crystallinity of WO_x . c, SAED pattern of YSZ.

The following texts were also added in the revised manuscript (Page 3)

“The YSZ and WO_x layers were partially crystallized during fabrication process which could contribute to better ion conductivity for switching (see Methods and Supplementary Fig. 1).”

2. In fig. 2, it seems that the computing was only shown for multiplication, rather than multiply-accumulate operation that has been demonstrated in RRAM arrays. Would the author comment on that?

RESPONSE: Thanks for the comment. The multiplication experiments were intended to demonstrate accurate analog-input-analog-weight multiplications between different pairs of device conductance and input values. We agree with the reviewer that multiply-accumulate operations are more important for in-memory computing and may be the better data to be presented in the manuscript. Therefore, we added experimental data of multiply-accumulate operations as figure 2f and 2g on page 5.

Fig. 2 Characterizations of the ECRAM. *f*, Experimental VMM output versus expected values in the 10×10 ECRAM array from 1,000 random input voltage vectors. *g*, Histogram plot of the output error from the VMM.

The following paragraph was also added in the revised manuscript (page 7)

“Vector-matrix multiplication (VMM) in a 10×10 ECRAM array was also evaluated. During read and VMM operations, input voltage pulses were applied at the source terminals through the select line electrodes, and the current were readout at the bit line outputs (Supplementary Fig. 8). 1,000 randomly-generated binary input voltage vectors were applied serially to the ECRAM array with randomly-initiated device conductance. The experimental VMM results were in a good agreement with the expected results (Fig. 2f) and demonstrated low computing errors (Fig. 2g).”

3. The authors showed that the device can achieve linear update in several different conductance ranges, while the retention data was only tested for 20-60 μS . It may be worth knowing how the device retention looks like in other switching windows, such as sub- μS and over 200 μS ?

RESPONSE: Thanks for the nice suggestion. Following this suggestion, we performed additional retention tests over a wide switching window ranging from sub- μS to over 200 μS and replaced the fig. 2d with the new retention data.

Fig. 2d, Retention test of 20 analog states over a wide switching window.

4. The open loop programming results are very impressive. Usually in RRAM arrays there are always some defective devices. I understand that ECRAM is non filamentary switching which means it may be more reliable, but can author comment on the reproducibility of the reported ECRAM?

RESPONSE: Thanks for the comment. Based on our experiences with RRAMs and ECRAMs, the ECRAM shows noticeably better reproducibility in two aspects. First, we did not observe ‘stuck-on’ and ‘stuck-off’ devices in ECRAM array so that all devices are operational. Second, although some switching variations were still observed, the overall uniformity is good, as confirmed by results from array operations. We agree with the reviewer that non filamentary switching may have played an important role in determining the reproducibility of the ECRAM.

5. The authors have only shown schematic of device fabrication, it would be better to show how the array was fabricated.

RESPONSE: We thank the reviewer for this constructive comment. We added the schematic of array fabrication process as Supplementary Fig. 11 to SI and revised the fabrication process in Methods section as shown below.

Supplementary Figure 11. Fabrication process of an ECRAM array. Left figure shows the layout of the ECRAM array, and the fabrication process of a unit cell is illustrated in right figure, including: **a**, selectline: Ti (3 nm) / Pt (40 nm), **b**, SiO₂ isolation layer: PECVD SiO₂ (100 nm), **c**, SiO₂ dry etch for selectline via and testing pads exposure, **d**, Source contacts (connected through selectline vias) and bitlines: Ti (3 nm) / Pt (40 nm) / W (5 nm), **e**, WO_x (100 nm) channels: reactive sputtering from W target with an Ar/O₂ ratio of 4: 3, followed with rapid thermal process (RTP) at 400 °C for 30 seconds, **f**, YSZ (50 nm) electrolyte layers: RF sputtering with 8 wt. % yttria-stabilized-zirconia target, **g**, Word lines (gate): W (50 nm).

ECRAM Array Fabrication

“ECRAM arrays were fabricated on top of a silicon substrate with a 100 nm thick SiO₂ layer. The fabrication process is schematically illustrated in Supplementary Fig. 11. First, Ti (3 nm) / Pt (40 nm) selectlines were formed by photolithography and DC sputtering. A 100 nm SiO₂ layer was then deposited using PECVD to provide isolation between bitlines and selectlines, followed by photolithography and ICP-RIE process to selectively expose bottom selectlines electrodes for electric contact to source terminals. Ti (3 nm) / Pt (40 nm) / W (5 nm) stack bitlines and source contacts were formed at the same time by photolithography and DC sputtering. After that, 100 nm thick WO_x channel were patterned and deposited using reactive DC sputtering with a W target under the Ar / O₂ (4:3) mixing atmosphere. The sputtering power density was set to 4.5 W/cm². A rapid thermal process at 400 °C for 30 seconds was subsequently conducted for modulating the crystallinity and electrical properties of obtained WO_x channels. 50 nm thick 8 wt. % YSZ electrolyte layer was further formed by RF sputtering under Ar ambient. Both WO_x and YSZ layer were patterned by photolithography for good electrical isolation between different cells in the array. Finally, wordlines of 50 nm tungsten were formed by photolithography and DC sputtering to complete the array fabrication process.”

6. There is a typo in the caption of fig. 5, probably some missing words from editing.

RESPONSE: We apologize for the mistake. We have carefully proofread the entire manuscript again to correct any grammatical errors in the revised manuscript.

Response to Reviewer #2:

The submitted manuscript provides a demonstration of the unique properties and potential of Electrochemical Random-Access Memories (ECRAM) as artificial synapses for in-memory compute. However, this reviewer requires further details and clarifications to properly gauge the novelty and impact of this systematic work. If you could please help understand and address, where applicable, the following.

RESPONSE: We thank the reviewer for the enlightening comments. Accordingly, we have conducted additional experiments and analysis, and thoroughly revised our manuscript. Our point-to-point responses to each question are as follows.

1. The manuscript puts an emphasis on "open-loop" programming and claims it is superior to the burden added by a write-verify scheme. That is only truly differentiating if it is combined with massively parallel and non-serialized programming. Is this report using a parallel programming scheme in the array demonstrators where all devices are updated simultaneously? If yes, please provide the details of such a scheme, and a plot showing how each device in an array can be programmed in such scheme with limited impact associated with half-select of its nearest neighbors.

RESPONSE: We thank the reviewer for this insightful comment. In this work, the ECRAM devices in the array was still updated in series using 1/2V biasing scheme, rather than fully parallel operations. The detailed description of array programming was

added in the Methods section. We agree with the reviewer that it will be “truly differentiating” if full array parallel update can be achieved. In fact, the pioneer study regarding the parallel programming in RPU arrays was one of the main motivations for our ECRAM study. The intent of the present manuscript was to show that good switching properties of ECRAM can be reproduced at array-level with relative uniform electrical behaviors across entire array, whereas to implement full array operations requires additional peripheral circuits and corresponding upper-level designs that are beyond the scope of this work. The demonstrated open-loop programming capability will be important for full array parallel programming, which we would like to explore in future work.

To address this great point from the reviewer, we added the following sentences in the manuscript (page 2).

“For programming-intensive tasks such as training neural networks, closed-loop operation will not only be inefficient, but could also limit the use of array-level parallel programming schemes³².”

We also agree with the reviewer on the notion that crosstalk should be limited for full array parallel operations. In our serial programming experiments, we used $1/2V$ biasing scheme (Supplementary Fig. 13a, as shown below) and found very minor impact on the half-selected cells in our 10×10 arrays, thanks to the nonlinear relationship between conductance change and programming voltages. However, the crosstalk will eventually be an issue in larger arrays. Following reviewer’s comments, we performed additional studies on the issue. In the revised manuscript, we measured crosstalk effect in half-selected cells (Supplementary Fig. 13b and 13c, as shown below) with derived average write disturbance of 1.8% to half-selected cells compared to selected devices.

Supplementary Figure 13. Crosstalk disturbance in half-selected ECRAM devices. *a*, Schematic of $1/2V$ programming scheme, *b*, Conductance update of four nearest devices in the array. *c*, Mean and standard deviation of relative conductance change in non-selected and half-selected devices compared to selected devices.

We simulated the pattern programming task accounting for crosstalk in different sizes of ECRAM arrays (Supplementary Fig. 14, as shown below). Simulation shows that open-loop programming becomes inaccurate in larger arrays due to accumulated impacts from half-selected cells. Nevertheless, it is possible to reduce programming errors in larger array (e.g. 128×128) by a few feedback adjustments (fig. b), this is an encouraging observation suggesting that larger arrays may still be used for iterative training tasks.

Supplementary Figure 14. Simulation to evaluate the impact of half-selected disturbance in larger arrays. *a*, Open-loop programming of down-sampled Lenna images into different sizes of ECRAM array. Half-selected devices were also updated in each programming cycle accounting for crosstalk disturbance. A half-voltage disturbance value of 3.0% was used to simulate the programming accuracy under slightly more challenging conditions than our experimental conditions. *b*, Iterative closed-loop programming in 128×128 array for 3 and 5 cycles, showing convergence of programming in larger arrays in the appearance of crosstalk interferences. For all simulations, the matrix started from randomly initialized values and each device was updated in series. The simulation code was programmed using MATLAB and intended as a coarse evaluation of the crosstalk effect. The simulation did not include the impact from line resistance and peripheral circuitry.

The detailed descriptions and schematic of 1/2V biasing scheme were added as Supplementary Fig. 13a (shown above) and in Methods section (Page 13) as follows.

“For array programming, the ECRAM devices were updated in series using 1/2V scheme to mitigate crosstalk in the array, utilizing the nonlinear relationship between the conductance change and programming voltage amplitudes (Supplementary Fig. 13 and 14).”

2. The manuscript demonstrates a 10x10 array with impressive programming range, including at the relatively high channel resistance necessary to scale to larger array while maintaining signal to noise. However, there is no diagram or clear description of the metal lines topology. In particular, the "wordlines" are said to be deposited subsequently to electrolytes. Are the active areas of devices defined in this case, or are the WO_x & YSZ deposited as blankets? If the latter, how can you electrically isolate devices if WO_x is not patterned?

RESPONSE: We thank the reviewer for bringing up this issue. The WO_x and YSZ layers were all isolated patterns defined by individual photolithography processes so that the active areas were all electrically isolated in our array design. We apologize for the lack of details regarding this matter and we have added schematic diagram of our fabrication process as Supplementary Fig. 11 to SI and revised the methods section as follows:

Supplementary Figure 11. Fabrication process of an ECRAM array. Left figure shows the layout of the ECRAM array, and the fabrication process of a unit cell is illustrated in right figure, including: **a**, selectline: Ti (3 nm) / Pt (40 nm), **b**, SiO₂ isolation layer: PECVD SiO₂ (100 nm), **c**, SiO₂ dry etch for selectline via and testing pads exposure, **d**, Source contacts (connected through selectline vias) and bitlines: Ti (3 nm) / Pt (40 nm) / W (5 nm), **e**, WO_x (100 nm) channels: reactive sputtering from W target with an Ar/O₂ ratio of 4: 3, followed with rapid thermal process (RTP) at 400 °C for 30 seconds, **f**, YSZ (50 nm) electrolyte layers: RF sputtering with 8 wt. % yttria-stabilized-zirconia target, **g**, Word lines (gate): W (50 nm).

ECRAM Array Fabrication

“ECRAM arrays were fabricated on top of a silicon substrate with a 100 nm thick SiO₂ layer. The fabrication process is schematically illustrated in Supplementary Fig. 11. First, Ti (3 nm) / Pt (40 nm) selectlines were formed by photolithography and DC sputtering. A 100 nm SiO₂ layer was then deposited using PECVD to provide isolation between bitlines and selectlines, followed by photolithography and ICP-RIE process to selectively expose bottom selectlines electrodes for electric contact to source terminals. Ti (3 nm) / Pt (40 nm) / W (5 nm) stack bitlines and source contacts were formed at the same time by photolithography and DC sputtering. After that, 100 nm thick WO_x channel were patterned and deposited using reactive DC sputtering with a W target under the Ar / O₂ (4:3) mixing atmosphere. The sputtering power density was set to 4.5 W/cm². A rapid thermal process at 400 °C for 30 seconds was subsequently conducted for modulating the crystallinity and electrical properties of obtained WO_x channels. 50 nm thick 8 wt. % YSZ electrolyte layer was further formed by RF sputtering under Ar ambient. **Both WO_x and YSZ layer were patterned by photolithography for good electrical isolation between different cells in the array.** Finally, wordlines of 50 nm tungsten were formed by photolithography and DC sputtering to complete the array fabrication process.”

3. The manuscript repeatedly mention the need for linear characteristics, but for deep learning applications, that is not a requirement, what should be emphasized is the symmetry allowing for algorithms to converge when used in learning tasks. In addition, it should be stated that modifying training algorithms can relax the metrics a discrete device needs to achieve. (see Gokmen and Haensch, Algorithm for Training Neural Networks on Resistive Device Arrays, Front. Neurosci., 26 February 2020)

RESPONSE: Thanks for the insightful comments and great advices on the device needs in learning tasks, as well as on the notion that modifying training algorithms can relax the requirements for hardware. We agree with reviewer that a main application of ECRAM would be for training neural network where the update symmetry is critical. In this work, we focused on developing a functional ECRAM array that can reproduce the promising properties of single ECRAM device to array-scale, and thus we hoped that we could be more comprehensive in optimizing the ECRAM performance in arrays towards the need of an “ideal” device.

Nevertheless, we realized that it is important to be accurate on the statement regarding the device requirements for learning tasks. Following the reviewer’s comment, we revised the manuscript to clarify the importance of symmetry and the possibility of relaxing hardware requirement by training algorithms, while also avoiding the overselling of linearity. The changes are listed as follows.

1) We added the following sentences in the revised manuscript on page 5:

*“However, it was found that maintaining **switching symmetry is sufficient** to guarantee good convergence in training neural networks while linearity mainly affects the effective learning rate of each weight update^{18,32,56}. Meanwhile, the metrics of a discrete device could be further **relaxed by modifying training algorithms**³².”*

2) We added related references in the revised manuscript.

32. Gokmen, T. & Haensch, W. Algorithm for Training Neural Networks on Resistive Device Arrays. *Front Neurosci* **14**, 103 (2020).

56. Gokmen, T. & Vlasov, Y. Acceleration of Deep Neural Network Training with Resistive Cross-Point Devices: Design Considerations. *Front Neurosci* **10**, 333 (2016).

3) We removed the claim of linearity for training larger neural network.

Original: *“It suggests that the uniform and linear ECRAM developed in this work could be used to accelerate the training process of larger neural networks”*

Revised: *“suggesting that securing uniformity and reducing outlier devices may be important in the training process of larger neural networks.”*

4) We added symmetry to claims regarding programming on page 2:

*“Ideally, analog programming should be an open-loop, linear **and symmetric** process.”*

*“ferroelectric memory device (FeRAM) has demonstrated good linear and symmetric programming capability⁴³, but switching linearity **and symmetry** is achieved by using voltage pulses with incremental amplitudes”*

4. A series of demonstrations are provided using the array: pattern transfer, weight transfer, and "in-situ" training. A general observation is that for inference tasks (such as the first 2 demonstrations) the use of ECRAM can hardly be justified over established technologies such as PCM, RRAM, Flash, etc... since you would transfer information once and then spend the vast majority of time and energy performing matrix-vector multiplication, making use of serial write-verify acceptable. Moreover, for the weight transfer demonstration, the network can be trained in software using "hardware-aware" noise injection to make it more resilient to known devices non-idealities and degradation. Because of the unique deterministic and controllable properties of ECRAM, it is the recommendation of this reviewer to focus on the training demonstration. Indeed, this is where it provides true differentiation. This could also allow the authors to provide more details on that particular achievement.

RESPONSE: We thank the reviewer for this constructive comment. It is also our intention to use ECRAM for training tasks rather than inference tasks. Given the size limit of our current ECRAM array, we were unable to perform tasks that are significantly more complex than the current demonstration. While we will try to make larger arrays for a full-scale investigation of the ECRAM in future work, here we tried to use the pattern programming tasks as a metric to gauge the accuracy of each weight update and complement the *in-situ* training task in evaluating the capability of this ECRAM array for training. For example, the tasks shown in figure 3 was designed to evaluate programming in iterative tasks like training. The motivation of this experiment was described on page 7 as:

“For training neural networks, the weights are directly stored in memory arrays and updated through repeated training cycles. The accuracy of the neural network is determined collectively from the update precision of each device. Therefore, reproducible analog programming from an entire array is highly important. Fig. 3 shows a series of open-loop programming tasks performed in our ECRAM array.”

Following the reviewer’s suggestions, we revised the manuscript to clarify the ambiguity of the task objectives.

We revised the paragraph in introduction section on Page 3 as:

*“In this article, we reported an electrochemical memory array with accurate open-loop analog programmability. Linear and symmetric conductance update of ECRAM was faithfully reproduced in integrated arrays with successful demonstrations in a set of image programming tasks, **showing promising programmability for accurate weight update operations**. We further experimentally employed our ECRAM arrays for training tasks and in-situ trained a bi-layer neural network to detect poisonous mushrooms with a software-like classification accuracy of 99.4%.”*

We added following illustrations in the revised manuscript (Page 9).

“Combining the in-memory computing capability demonstrated in Fig. 2 and reproducible analog programming capability in Fig. 3, our ECRAM array provides a promising toolset for in-situ training tasks”

5. It appears that the favored programming voltages and pulse durations are +/-5V and 100ms, respectively. What are the thoughts of the authors on how to reduce both numbers, especially pulse width, without degrading signal to noise ratio?

RESPONSE: The speed of our ECRAM is indeed a concern for us which requires further optimizations. A few recent papers have shed some lights on this topic (*Kim, et al. IEDM 2019; Onen, M. et al. Science 2022; Cui, et al. Nature Electronics 2023*). A general observation is that the switching speed of ECRAMs is dominated by the ion conductivity in electrolyte and ion intercalation/de-intercalation in channels. The table below summarize the approaches to achieve fast switching of ECRAM in their papers.

#	Pulse width	Channel/ electrolyte materials	Mobile ions	Approaches for high speed	Reference
1	< 1 μ s	PEDOT: PSS/ Nafion	H ⁺	Employing Nafion with high ion conductivity of hydrogen for fast ion movement.	E. Fuller et al. Science 364, 570-574, 2019
2	2 μ s	TiO ₂ / YSZ	O ²⁻	Using external heating to lower energy barrier for fast oxygen ions movement.	Y. Li et al. Adv. Mater. 32, e2003984, 2020
3	5 ns	WO ₃ / PSG	H ⁺	Adopting ultrathin PSG as electrolyte to introduce high electric field across the gate/electrolyte/channel stack.	M. Onen et al. Science 377, 539-543, 2022
4	5 μ s	WO ₃ / ZrO ₂	H ⁺	Engineering amorphous and stoichiometric WO ₃ channels for easier intercalation/de-intercalation.	J. Cui et al. Nat. Electron. 6, 292-300, 2023

Following these studies, there are some possible solutions to improve the speed of our ECRAM, which include: 1) improving the ion conductivity of electrolyte by using alternative electrolyte materials and/or modulation of material stoichiometry. 2) tuning channels for easier ions intercalation/de-intercalation such as modulation of metal/oxygen ratio and/or engineering the channel/electrolyte interface. 3) developing reproducible methods to make thinner and robust devices.

On page 11 of the main text, we added a short discussion as follows.

“The efficacy of ECRAM may be further improved in future studies by optimizing switching speed and reducing outlier devices in large-scale integrations.”

We added related references in the revised manuscript.

48. Onen, M. et al. Nanosecond protonic programmable resistors for analog deep learning. *Science* 377, 539-543 (2022).

49. Li, Y.Y. & Chueh, W.C. Electrochemical and Chemical Insertion for Energy Transformation and Switching. *Annu. Rev. Mater. Res.*, 48, 137-165 (2018).

6. What is the authors' expectation on the impact of a BEOL integration process environment on the array properties during steps following active area formation? (exposure to further thermal budget, chemistry, ionic species, moisture, etc...)

RESPONSE: Among all treatments the ECRAM might go through in the BEOL integration process, we have major concerns over the thermal treatment. In our experiences, the device could degrade after exposure to thermal budget over 300 °C. Although the BEOL integration process might need to be tailored for the ECRAM integration, a practical strategy for us at the moment is to fabricate ECRAM on top of metal layers to avoid any excessive thermal and chemical exposures during the BEOL process, which has been adopted for some RRAM-based integrated chip (*Li, et al. Nat. Commun. 2018, Rao, et al. Nature 2023*).

This reviewer hopes that the above will be of use to the authors and can be used to further amplify the systematic study of their ECRAM devices + arrays, and complement the unique demonstrations presented. John Rozen

RESPONSE: We would like to express our sincere gratitude to the reviewer once again for the valuable and constructive suggestions, which we found to be educational and very helpful.

Response to Reviewer #3:

In the present manuscript, the authors disclose a study on the open-loop programming of electrochemical memory arrays. The authors demonstrate that the devices can be programmed highly accurately using a pulse-width modulation scheme. The devices show very linear and symmetric tuning which makes them suitable as weights in online learning systems using backpropagation. In addition, the authors demonstrate the integration of the device in a small array as well as the programming within the array. In-situ training is demonstrated on the array, mainly by programming the respective devices. Based on the device's properties a large-scale network is simulated using standard software. ECRAMs have been studied by different groups. Also, small arrays have been demonstrated (e.g., by Cui et al, Nature 2023, ref. 47 in the manuscript). Thus, the device concept itself is not novel, also the realization of an integrated array has been demonstrated before. The novelty is shown in the programming scheme. On the experimental basis shown in the manuscript, one cannot draw the conclusion of superior properties. A lot of information is missing to be able to reproduce the results by the authors. The manuscript requires extensive revisions according to the comments outlined below.

RESPONSE: We thank the reviewer for summarizing the strength and weakness of our work and offering constructive suggestions that have helped us to improve our manuscript. Following these suggestions, we have revised the manuscript extensively with new experimental data and analysis. We hope that we have addressed the reviewer's concerns in the revised manuscript.

1) While the programming scheme and properties look very promising the study does not prove that the integration of large arrays with low variability is feasible. The authors

demonstrate here a very low d2d variability for 10 devices over 5 cycles. For the network shown in Fig. 5, however, a lot higher number of devices is required, and the devices will be trained for more than 5 cycles. Even though most of the devices work well, the outliers will limit the overall performance. This might be 1 out of 1000 or even less devices, but for large scale networks this needs to be guaranteed. Such an investigation is certainly out of the scope of this paper, but one should be careful to claim superior properties with such a low number of statistics.

RESPONSE: Thanks for bringing up this issue. We agree with reviewer that measuring small number of the devices is not sufficient to validate their feasibility in large-scale integrations. Therefore, we have toned down the claim regarding uniformity and the performance of ECRAM in large-scale arrays. The list of changes in the revised manuscript are as follows:

Page #	Original Manuscript	Revised Manuscript	Changes
Page 3	Linear and symmetric conductance update was achieved in integrated arrays with excellent spatial and temporal uniformity	Linear and symmetric conductance update of ECRAM was faithfully reproduced in integrated arrays	Removed claim of “Excellent uniformity”
Page 3	simulation based on device characteristics showed that the ECRAM arrays can achieve highly accurate training of large neural networks using open loop analog programming and demonstrating software comparable accuracy	simulation based on device characteristics showed that the ECRAM arrays can achieve highly accurate training of large neural networks such as VGG-8	Removed claim of achieving software comparable accuracy in large network
Page 4 subtitle	Uniform ECRAM device characteristics	ECRAM characteristics	Removed “Uniform”
Page 5	Fig. 2c shows the cumulative probability curve of the result from Fig. 2b with conductance data extracted from every two pulses. Tight and uniform distribution of these 25 states were observed among different devices and up/down cycles.		Removed Fig. 2c and corresponding descriptions in main text regarding uniformity analysis
Page 11	It suggests that the uniform and linear ECRAM developed in this work could be used to accelerate the training process of larger neural networks.	suggesting that securing uniformity and reducing outlier devices may be important in the training process of larger neural networks.	Removed claim of uniformity in large network and pointed out outliers

Page 11	Open-loop analog programmability in array-level demonstrations was achieved utilizing the intrinsic device properties such as excellent spatial-temporal uniformity	Open-loop analog programmability in array-level demonstrations was achieved utilizing the intrinsic device properties such as good spatial-temporal uniformity	Replaced “excellent” to “good”
Page 11	The excellent uniformity of our ECRAM have contributed to software-comparable training accuracy for neural networks	The demonstrated analog programmability of our ECRAM have contributed to software-comparable training accuracy for neural networks	Removed “excellent uniformity”

2) What is the endurance of the devices? Some thousands of cycles are shown, but what is the limit?

RESPONSE: Thanks for the comment. In the revised manuscript, we’ve added endurance test as Supplementary Fig. 7. Here we used $\pm 5V/1ms$ pulse and measured up to 50 million pulses. Slight degradation in the switching linearity was observed, but the overall switching behavior under same pulse conditions remained consistent, showing good endurance. Currently, the extent of endurance test was mainly limited by the relative slow programming speed of the ECRAM, which will be a priority for us to improve in near future.

Supplementary Figure 7. Endurance test of the device. Pulse condition: $V_G = \pm 5V$, $t_{pulse} = 1ms$.

3) For the programming the authors use pulses of 5V and 150 ms. The voltage is too high to be integrated with modern CMOS. Thus, special transistors and drivers would need to be used creating circuitry overhead and additional power consumption. The pulses are quite long with 150 ms, leading to slow speed operation and a higher energy consumption.

RESPONSE: We agree with reviewer that the speed and programming voltage is a concern for our ECRAM device. In fact, the speed has been a common issue for ECRAM in many prior publications. Recently, we see some encouraging reports from colleagues (*Kim, et al. IEDM 2019, Onen, M. et al. Science 2022, Cui, et al. Nature Electronics 2023*), which have showcased the feasibility in improving the switching speed of ECRAM. A general observation is that the switching speed of ECRAMs is dominated by the ion conductivity in electrolyte and ion intercalation/de-intercalation in channels. The table below summarize the approaches to achieve fast switching of ECRAM in their papers.

#	Pulse width	Channel/ electrolyte materials	Mobile ions	Approaches for high speed	Reference
1	< 1 μ s	PEDOT: PSS/ Nafion	H ⁺	Employing Nafion with high ion conductivity of hydrogen for fast ion movement.	E. Fuller et al. Science 364, 570-574, 2019
2	2 μ s	TiO ₂ / YSZ	O ²⁻	Using external heating to lower energy barrier for fast oxygen ions movement.	Y. Li et al. Adv. Mater. 32, e2003984, 2020
3	5 ns	WO ₃ / PSG	H ⁺	Adopting ultrathin PSG as electrolyte to introduce high electric field across the gate/electrolyte/channel stack.	M. Onen et al. Science 377, 539-543, 2022
4	5 μ s	WO ₃ / ZrO ₂	H ⁺	Engineering amorphous and stoichiometric WO ₃ channels for easier intercalation/de-intercalation.	J. Cui et al. Nat. Electron. 6, 292–300, 2023

Following these studies, there are some possible solutions to improve the speed of our ECRAM, which include: 1) improving the ion conductivity of electrolyte by using alternative electrolyte materials and/or modulation of material stoichiometry. 2) tuning channels for easier ions intercalation/de-intercalation such as modulation of metal/oxygen ratio and/or engineering the channel/electrolyte interface. 3) developing reproducible methods to make thinner and robust devices.

On page 12 of the main text, we added a short discussion as follows:

“The efficacy of ECRAM may be further improved in future studies by optimizing switching speed and reducing outlier devices in large-scale integrations.”

We added related references in the revised manuscript.

48. Onen, M. et al. Nanosecond protonic programmable resistors for analog deep learning. *Science* 377, 539-543 (2022).

49. Li, Y.Y. & Chueh, W.C. Electrochemical and Chemical Insertion for Energy Transformation and Switching. *Annu. Rev. Mater. Res.*, 48, 137-165 (2018).

4) The programming of the device in the array is not well explained. The ECRAM is a three-terminal device. The lines in the array are not labeled. What are the gate lines, and which are the drain and source lines? According to the supplement, the authors connect all pins of the array to a probe card. What signals are applied to all lines when programming? Do you program one cell at a time? How is program-disturb avoided in the other cells sharing the same gate line?

RESPONSE: Thanks for this comment. We apologies for these missing details. In the revised manuscript, we have added new data and analysis for the programming process, which includes:

(1) We added detailed descriptions regarding array programming in Methods section.

“For array programming, the ECRAM devices were updated in series using 1/2V scheme to mitigate crosstalk in the array, utilizing the nonlinear relationship between the conductance change and programming voltage amplitudes (Supplementary Fig. 13 and 14). Single pulses with variable pulse durations were used to change devices from one conductance state to the other.”

“For open-loop programming, a single programming pulse was applied to each selected device without feedback, while for closed-loop programming, the device was programmed using iterative read-program-read cycles.”

(2) We added Supplementary Figure 13a to SI, which shows the detailed labeling of word/gate line, select/source line and bit/drain line, and the biasing for each line to implement the 1/2V programming scheme. In this work, each cell in the array was programmed in series.

Supplementary Figure 13. Crosstalk disturbance in half-selected ECRAM devices. *a*, Schematic of 1/2V programming scheme, *b*, Conductance update of four nearest devices in the array. *c*, Relative conductance change of non-selected and half-selected devices in comparison to selected devices.

(3) In our serial programming experiments, we found very minor impact on the half-selected cells in our 10×10 arrays when using 1/2V biasing scheme, thanks to the nonlinear relationship between conductance change and programming voltages. In Supplementary Figure 13b and c, we added experimental tests to evaluate the program-disturbance to half-selected cells, as shown above, with 1.8% relative conductance change in half-selected cells compared to selected devices.

(4) We added Supplementary Fig. 14 to show the simulation of pattern programming task accounting for crosstalk in different sizes of ECRAM arrays, as shown below. It shows that open-loop programming becomes inaccurate in larger arrays due to accumulated impacts from half-selected cells, while it is also possible to reduce programming errors in larger array (e.g. 128×128) by feedback adjustments (fig. b).

Supplementary Figure 14. Simulation of half-selected disturbance in larger arrays. *a*, Open-loop programming of down-sampled Lenna images into different sizes of ECRAM array. Half-selected devices were also updated in each programming cycle accounting for crosstalk disturbance. A half-voltage disturbance value of 3.0% was used to simulate the programming accuracy under slightly more challenging conditions than our experimental conditions. *b*, Iterative closed-loop programming in 128×128 array for 3 and 5 cycles, showing convergence of programming in larger arrays in the appearance of crosstalk interferences. For all simulations, the matrix started from randomly initialized values and each device was updated in series. The simulation code was programmed using MATLAB and intended as a coarse evaluation of the crosstalk effect. The simulation did not include the impact from line resistance and peripheral circuitry.

5) What is the read voltage used? How is the read performed in the array, when reading a single cell and a whole line for doing the VMM operation?

RESPONSE: We used 0.5 V for reading in the array. To illustrate the read process, we added a diagram as *Supplementary Figure 8* to SI. For reading operations, we grounded the word lines (WL), as well as all unselected bit lines (BL) and select lines (SL). The read voltages were applied at the SLs while current was readout at selected BL. For reading a single cell, a V_{read} pulse was applied only to the SL of the target cell, while for VMM, binary voltage vectors are applied to all SLs.

Supplementary Figure 8. Schematic diagrams of the read operations in an array. *a*, Schematic of read process of a single device, *b*, Schematic of read process of a whole line for VMM operations. During read operation, all word lines and unselected source/drain lines are grounded. Read pulses are applied to selected source lines, then record the output current along the selected drain line.

On page 7 of the main text, we added the following information.

“During read and VMM operations, input voltage pulses were applied at the source terminals through the select line electrodes, and the current were readout at the bit line outputs (Supplementary Fig. 8)”

We also added biasing scheme during reading in the Methods.

“For reading operations, the word lines (WL), as well as all unselected bit lines (BL) and select lines (SL) were grounded (Supplementary Figure 8). Read voltages of 0.5V were applied at the SLs while current was readout at selected BL.”

6) The multiplication test in 2f is not well explained. In general, one should program random patterns to the devices and use random inputs, but it is not clear if this has been done. Here, it is also helpful to discuss the addressing.

RESPONSE: Thank the reviewer for the valuable comments. In original Fig. 2f, we used fixed-valued input and weight vectors to perform analog-input-analog-weight multiplications (device conductance from $31\mu\text{S}$ to $39\mu\text{S}$, while input from 0.1V to 0.9V). Column output was readout for each multiplication individually to gauge the multiplication accuracy. Following reviewer’s suggestion, we did VMM tests with randomly generated pattern and inputs. 1,000 VMM results were statistically analyzed and added to Figure 2 as Fig. 2f and 2g on page 5 of the revised manuscript. Meanwhile, the addressing of the array for VMM was added as Supplementary Fig. 8 to SI (as shown in response to previous question).

Fig. 2 Characterizations of the ECRAM. *f*, Experimental VMM output versus expected values in the 10×10 ECRAM array from 1,000 random input voltage vectors. *g*, Histogram plot of the output error from the VMM.

The following paragraph was also added in the revised manuscript (page 7)

“Vector-matrix multiplication (VMM) in a 10×10 ECRAM array was also evaluated. During read and VMM operations, input voltage pulses were applied at the source terminals through the select line electrodes, and the current were readout at the bit line outputs (Supplementary Fig. 8). 1,000 randomly-generated binary input voltage vectors were applied serially to the ECRAM array with randomly-initiated device conductance. The experimental VMM results were in a good agreement with the expected results (Fig. 2f) and demonstrated low computing errors (Fig. 2g).”

7) On page 6, the authors mention that a voltage sweep is used to read the device state. What was the used sweep rate?

RESPONSE: We used sweep rate of 0.8V/s for the voltage sweep to read the device state. We added the following sentences in Methods sections as follows:

“For reading the device’s conductance using DC sweep, we used sweep voltages from 0 to 1V across the source and drain electrodes at a sweep rate of 0.8V/s.”

8) For the demonstration of the two-layer network, the authors store the weights for both layers in the same array. It is not clear how inference and backpropagation is realized here. How is the output of the first layer introduced in the 2nd layer? As there is no additional circuitry, the authors must do this in software, but the explanation is lacking. Which activation function is used between the first and second layer? The output of the VMM in the first layer is current, which needs to be transferred to a reading voltage pattern. The reading needs to be done sequentially here.

RESPONSE: Thanks for bringing up the issue. The reviewer is right that part of the computation for this demonstration was done in software. Our current hardware system does not implement hardware circuits for activation functions and data routing. Therefore, the processing of activation function for both layers and the data communications between two layers were all done in software. In the revised manuscript, we have extensively revised the relative section in Methods to include detailed descriptions regarding the training demonstration as follows.

On page 9 of the revised manuscript, we provide clarification regarding the software part of implementation used in this demonstration.

“For hardware implementation, only synaptic functions have been implemented in arrays, while the processing of activation function, routing data between first and second neural network layers, as well as the calculation of gradients for back-propagation were done in software.”

We also added the following sentences to Methods on Page 14:

“For bilayer neural network training in ECRAM array, we employed softmax activation for both the first and second layers, and the activations were implemented in software.”

“After the forward process in each epoch, the neural network loss and weight gradients were calculated in software, and linearly mapped to required conductance change.”

9) In Fig. 4d, the accuracy of the network is demonstrated, but a saturation has not been achieved. Why did the authors stop after about 38 epochs. What happens further on? The authors should train until a saturation is achieved as for example in Fig. 5. For benchmarking it would be good to show the accuracy that is expected using a pure software approach.

RESPONSE: We appreciate the reviewer for this valuable suggestion. It was a mistake for us to accidentally stop the training. Following reviewer’s suggestion, we performed the in-situ training again and optimized the training parameters such as the learning rate.

We stopped the training after 70 epochs where saturation was achieved. By conducting further training, we were able to bring the classification accuracy to software-like accuracy of 99.4%, which was significantly better than that from original manuscript.

The related contents in Fig. 4 have been replaced with the new results (Page 10).

Fig 4. In-situ training using ECRAM array. *d*, Accuracy evolution curves as a function of training epochs derived from hardware incorporated neural networks (red), achieving classification accuracy of 99.4%, which matched the accuracy of pure software training (blue). *e*, Evolution of the conductance and *f*, programming errors of the 60 devices through 70 training epochs. *g*, Statistical analysis of programming error through training, the colored area shows the margin between $\mu-\sigma$ and $\mu+\sigma$.

We also replaced the accuracy number of 95% in original manuscript with the new results of 99.4% throughout the manuscript.

10) For the large-scale network the authors used a simulation tool. Which software has been used? Which device characteristics for the update have been used? Did the authors include the observe variability in the training? As mentioned in my first comment, it is not proven that the variability is as low for the high number of devices required for the simulated network. The used software also does not consider losses due to the CMOS circuitry and the analog to digital conversion.

RESPONSE: We thank the reviewer for the comment. In this work, we used the software framework of *DNN+NeuroSim* developed by Prof. Shimeng Yu's group at Gatech (Ref. 54 in the original manuscript) for benchmarking the performance of compute-in-memory technologies. For ECRAM simulation, we built behavior model of ECRAM which considered the device-to-device variation, cycle-to-cycle variation, nonlinearity, analog on/off ratio, conductance level, quantization of ADCs. We agree with reviewer that the simulation will not be fully accurate in characterizing the actual ECRAM array performance in training large neural network due to existence of outlier devices in larger arrays and circuit level non-idealities. In the revised manuscript, we tried to perform additional simulations accounting for increasing variations to provide a coarse evaluation of the training capabilities using the ECRAM array. The detailed changes in the revised manuscript are listed below:

1) The simulation results in Fig. 5 were replaced with new data considering different level of variations (Page 11).

Fig 5. Simulation of VGG-8 with CIFAR-10 database. *a*, The network structure of the VGG-8 network for CIFAR-10 image classification. *b*, Simulated training in ECRAM array with different level of device variability. *c*, The confusion matrix for trained VGG-8 on the test set.

2) Detailed information regarding the simulation was added to Methods section in the revised manuscript on Page 15.

Simulation of large neural network

“The simulation is conducted based on the Python framework of ‘DNN+NeuroSim’⁵⁸. The VGG-8 network consists of 3 convolutional blocks and 2 fully connected layers with total of 8 network layers, which was trained with CIFAR-10 dataset. After training with 50,000 images, the neural network is tested with 10,000 images for benchmarks of different devices models. Behavioral device model of ECRAM was built by fitting the experimental data shown in Fig. 2b. Cycle-to-cycle variation of 0.8% and device-to-device variation of 2.3% was used for the simulation. **To provide a coarse estimation of outlier devices in large-scale integrations, additional simulation with enlarged variation value of ($\times 1.2$, $\times 1.4$) was performed.** The large variation ratio was applied to both device-to-device and cycle-to-cycle variations, while keeping other parameters unchanged.”

3) Further clarification on the need to consider outlier devices in large-scale integrations was added to main text on Page 11.

“For simulation conditions with increased variability, gradual degradations in classification accuracies were observed, suggesting that securing uniformity and reducing outlier devices may be important in the training process of larger neural networks.”

REVIEWER COMMENTS

Reviewer #1 (Remarks to the Author):

The authors have addressed the questions from the reviewers. I would like to recommend the acceptance.

Reviewer #2 (Remarks to the Author):

This Reviewer would like to acknowledge the thorough effort made by the authors on answering all queries, amending the text & graphics, as well as performing further experiments.

Based on the added information, it appears the novelty of the work has two main axis;

- a. The demonstration of a relatively large array of ECRAM synaptic cells with appropriate conductance range and programmability with limited cross-talk
- b. The demonstration of an in-situ training task of a small-scale neural network highlighting the promise of ECRAM deterministic switching as part of analog accelerator fabrics

The main gaps of the work when it comes to bridging to a full-scale solution are recognized by the authors and only limited by the available measurement equipment, array fabrication capabilities, and the need for further material and device engineering;

- c. The demonstration do not include massively parallel weight updates without which in-memory analog training is highly inefficient
- d. The demonstration does not identify a path towards back-end of line integration which would require additional deposition and patterning steps likely to affect the material properties of the ECRAM array
- e. The demonstration does not indicate capability for high-speed switching of the devices but the authors do now address the steps that could address it

It is the opinion of this Reviewer that items #c and #d could be considered out of scope for this publication and exploratory work as long as it is mentioned in the text. Accordingly, this Reviewer

recommends publication of this systematic study pending further minor revisions listed below as items #a and #b do warrant this work to be shared with the community as it will certainly create further interest in these type of non-volatile memory element and applications in AI towards more sustainable and scalable computing.

Further need for revision:

1. The added array biasing scheme in Figure S13 does demonstrate limited half-select behavior but is not practical. Instead of using 0, $V/2$, V biases, the authors should use $-V/2$, 0, $V/2$ as 0V will be the default and most power efficient condition.

2. The added ECRAM fabrication details do clarify the topology of metal lines but fail to indicate the patterning method for the WO_x channel, YSZ gate/electrolyte and W electrodes (lift-off/RIE ?)

3. The table provided in answer to the topic of device speed and programming bias scalability interestingly does not include numbers from reference #36 (Kim et al.) which was the first to demonstrate a BEOL-compatible oxide-based solid-state ECRAM array, which was programmed with pulse widths down to 10ns. As a side note, this Reviewer would like to caution the authors on the interpretation of device speed across different publications if not normalized for a target conductance change and signal/noise ratio; it can indeed be quite arbitrary. In addition, the settling time needed post application of the write pulse and prior to reading the ECRAM cell is rarely mentioned and does matter in the overall achievable operating speed of the device.

4. The authors have addressed the question around consequences of further BEOL integration steps on the properties of the ECRAM cell but failed to add any such comment in the manuscript.

5. For the in-situ training task, please add details on the following two items:

Input parameters - what is the granularity of each parameter? i.e. how many colors? how many shapes?, how many sizes? etc.

Weight to conductance mapping - how was signal to noise optimized? i.e. how many levels/pulses are needed to cover the entire range for the parameters above and how was it chosen?

This Reviewer hopes that the above will be of use to the authors and can be leveraged to further amplify the systematic study of their ECRAM devices + arrays, and complement the unique demonstrations presented.

John Rozen

Reviewer #3 (Remarks to the Author):

All my comments have been addressed and I have no more questions. The work shows a nice demonstration on improving ECRAM technology. The CMOS compatibility, however, remains a challenge for this technology (compatibility of voltage and current levels). The authors should mention this in the conclusions as future challenge.

AUTHORS' RESPONSE TO THE REVIEWER'S COMMENTS

We thank the editor and reviewers again for their positive and highly valuable feedback on our manuscript. We have added additional information in the revised the manuscript.

Reviewer #1 (Remarks to the Author):

The authors have addressed the questions from the reviewers. I would like to recommend the acceptance.

RESPONSE: We sincerely thank the reviewer for the precious time to review our manuscript and offer constructive comments.

Reviewer #2 (Remarks to the Author):

This Reviewer would like to acknowledge the thorough effort made by the authors on answering all queries, amending the text & graphics, as well as performing further experiments.

Based on the added information, it appears the novelty of the work has two main axis;

- a. The demonstration of a relatively large array of ECRAM synaptic cells with appropriate conductance range and programmability with limited cross-talk.
- b. The demonstration of an in-situ training task of a small-scale neural network highlighting the promise of ECRAM deterministic switching as part of analog accelerator fabrics.

The main gaps of the work when it comes to bridging to a full-scale solution are recognized by the authors and only limited by the available measurement equipment, array fabrication capabilities, and the need for further material and device engineering;

- c. The demonstration do not include massively parallel weight updates without which in-memory analog training is highly inefficient
- d. The demonstration does not identify a path towards back-end of line integration which would require additional deposition and patterning steps likely to affect the material properties of the ECRAM array.
- e. The demonstration does not indicate capability for high-speed switching of the devices but the authors do now address the steps that could address it.

It is the opinion of this Reviewer that items #c and #d could be considered out of scope for this publication and exploratory work as long as it is mentioned in the text. Accordingly, this Reviewer recommends publication of this systematic study pending further minor revisions listed below as items #a and #b do warrant this work to be shared with the community as it will certainly create further interest in these type of non-volatile memory element and applications in AI towards more sustainable and scalable computing.

RESPONSE: We would like to express our sincere gratitude to the reviewer for the comprehensive remarks regarding the strengths and weakness of our manuscript. We have further revised our manuscript to address reviewer's suggestions.

Further need for revision:

1. The added array biasing scheme in Figure S13 does demonstrate limited half-select behavior but is not practical. Instead of using 0, $V/2$, V biases, the authors should use $-V/2$, 0, $V/2$ as 0V will be the default and most power efficient condition.

RESPONSE: Thanks for the comment. We agree with reviewer that using $-1/2V$, 0 and $V/2$ could be more efficient, but we have to use 0, $V/2$ and V biases in this work due to the inability of our current system to apply both positive and negative pulses at the same time. The 0, $V/2$ and V biases were also experimentally used in prior publications (e.g. F. Cai *et al*, Nat. Electron. 1, 137-145, 2018, I. Giannopoulos *et al*, Adv. Intell. Syst. 2, 2000141, 2020). We will explore the suggested implementations in future designs.

We acknowledged the issue in the caption of Figure S13 as follows.

“Supplementary Figure 13a., Schematic of $1/2V$ programming scheme, 0V, $V/2$ and V were used to achieve half-selection. A more efficient scheme would be $-1/2V$, 0V and $1/2V$ but was not supported by the current system.”

2. The added ECRAM fabrication details do clarify the topology of metal lines but fail to indicate the patterning method for the WO_x channel, YSZ gate/electrolyte and W electrodes (lift-off/RIE ?)

RESPONSE: We used lift-off for all patterned layers. We have added the following sentence in the Methods section.

“All patterns, including metal electrodes, WO_x channel and YSZ layers, were fabricated using lift-off process.”

3. The table provided in answer to the topic of device speed and programming bias scalability interestingly does not include numbers from reference #36 (Kim et al.) which was the first to demonstrate a BEOL-compatible oxide-based solid-state ECRAM array, which was programmed with pulse widths down to 10ns. As a side note, this Reviewer would like to caution the authors on the interpretation of device speed across different publications if not normalized for a target conductance change and signal/noise ratio; it can indeed be quite arbitrary. In addition, the settling time needed post application of the write pulse and prior to reading the ECRAM cell is rarely mentioned and does matter in the overall achievable operating speed of the device.

RESPONSE: Reference #36 by Kim *et al* was an important milestone in ECRAM research which we have followed and referenced in the manuscript. We did not include it in the table as this shortened IEDM report did not provide detailed discussions regarding its fast-switching mechanism. Even though we speculate strong electric field from its thin solid electrolyte layer may be an important factor for its nanosecond switching, we did not want to be inaccurate in specifying its mechanism in the table.

We thank the reviewer for offering additional insights in this matter and we have added the following discussions in the characterization section of the revised manuscript.

“The evaluation of the programming speed of the ECRAM should not only consider the pulsewidth of the write pulse, but also the amount of conductance change and

signal/noise ratio. The settling time between the write and read pulses, i.e. the read-after-write delay, should also be considered when evaluating the overall programming speed of ECRAM⁹. For the up/down test, the time interval between write and read pulses was set to 100 ms.”

59. Solomon, P.M. et al. Transient Investigation of Metal-oxide based, CMOS-compatible ECRAM in 2021 IEEE International Reliability Physics Symposium (IRPS), 1-7 (IEEE, 2021).

4. The authors have addressed the question around consequences of further BEOL integration steps on the properties of the ECRAM cell but failed to add any such comment in the manuscript.

RESPONSE: We added comments on BEOL integration steps in the Conclusion section of the revised manuscript as follows:

“The CMOS compatibility of the ECRAM array should also be addressed in integrated systems, ensuring consistent device performance undergoing BEOL fabrication process and low voltage operations matching CMOS designs in advanced technology nodes.”

We also added the following statements in the Methods section:

“The overall fabrication process was designed to avoid excessive thermal and chemical exposures that could degrade the device performance, which should be further optimized for the BEOL process to build an integrated ECRAM chip.”

5. For the in-situ training task, please add details on the following two items:

Input parameters - what is the granularity of each parameter? i.e. how many colors? how many shapes?, how many sizes? etc.

Weight to conductance mapping - how was signal to noise optimized? i.e. how many levels/pulses are needed to cover the entire range for the parameters above and how was it chosen?

RESPONSE: The **input parameters** including shape, size and others were added as Supplementary Table 1 and shown below. The description regarding the design of **weight to conductance mapping** was added to the Methods section. We linearly mapped the weight values between -1 to 1 to conductance ranges between 25 μS to 50 μS , which provides fine granularity for weight updates and good signal-to-noise ratio for inference with around 200 resolvable steps using a write pulse width of 100 ms, as shown in Fig. 2a (shown below). This conductance range was chosen empirically, which provided us with consistent training results for current task.

Fig. 2a. Linear and symmetric conductance update across different dynamic ranges of i) 0.1~0.2 μS , ii) 0.4~1.0 μS , iii) 2~4 μS , iv) 15~30 μS , v) 60~100 μS and vi) 170~230 μS , each cycle consists of 100 up/down voltage pulses (± 5 V/100 ms).

We added the following sentences in the Methods section of the revised manuscript:

“The first 10 attributes of the mushroom dataset were selected as the input parameters to the bilayer neural network, as summarized in Supplementary Table 1. Different descriptions of each input parameter were encoded numerically (e.g., different Cap-shapes were encoded from 0-5).”

Supplementary Table 1. Summary of input parameters for in-situ training tasks.

#	Input parameters	Descriptions
1	Cap-shape	bell, conical, convex, flat, knobbed, sunken
2	Cap-surface	fibrous, grooves, scaly, smooth
3	Cap-color	brown, buff, cinnamon, gray, green, pink, purple, red, white, yellow
4	Bruises	bruises, no
5	Odor	almond, anise, creosote, fishy, foul, musty, none, pungent, spicy
6	Gill-attachment	attached, descending, free, notched
7	Gill-spacing	close, crowded, distant
8	Gill-size	Broad, narrow
9	Gill-color	black, brown, buff, chocolate, gray, green, orange, pink, purple, red, white, yellow
10	Stalk-shape	enlarging, tapering

We also added the following sentences regarding weight to conductance mapping in the Method section of the revised manuscript:

“The weights (from -1 to 1) were linearly mapped to conductance ranges between 25 μ S to 50 μ S, which provides fine granularity for weight updates and good signal-to-noise ratio for inference tasks with around 200 resolvable steps using a write pulse width of 100 ms. This conductance range was chosen empirically, which provided us with consistent training results for current task.”

This Reviewer hopes that the above will be of use to the authors and can be leveraged to further amplify the systematic study of their ECRAM devices + arrays, and complement the unique demonstrations presented. John Rozen

RESPONSE: We would like to once again thank the reviewer for the constructive comments which have provided us with valuable knowledges and inspirations.

Reviewer #3 (Remarks to the Author):

All my comments have been addressed and I have no more questions. The work shows a nice demonstration on improving ECRAM technology. The CMOS compatibility, however, remains a challenge for this technology (compatibility of voltage and current levels). The authors should mention this in the conclusions as future challenge.

RESPONSE: We sincerely thank the reviewer for the precious time and valuable comments. To address reviewer’s suggestion, we have added the following sentences in the conclusion section.

“The CMOS compatibility of the ECRAM array should also be addressed in integrated systems, ensuring consistent device performance undergoing BEOL fabrication process and low voltage operations matching CMOS designs in advanced technology nodes.”

REVIEWERS' COMMENTS

Reviewer #2 (Remarks to the Author):

This Reviewer once more thanks the Authors for addressing all questions and suggestions and recommends publication of the revised manuscript which will certainly create further interest in these type of non-volatile memory element and applications in AI towards more sustainable and scalable computing.

AUTHORS' RESPONSE TO THE REVIEWER'S COMMENTS

Reviewer #2 (Remarks to the Author):

This Reviewer once more thanks the Authors for addressing all questions and suggestions and recommends publication of the revised manuscript which will certainly create further interest in these type of non-volatile memory element and applications in AI towards more sustainable and scalable computing.

RESPONSE: We are delighted that the reviewer recommends the publication of our manuscript and sincerely thank the reviewer for the insightful comments.